# Components of iron–Sulfur cluster assembly machineries are robust phylogenetic markers to trace the origin of mitochondria and plastids

**Pierre Simon Garcia**[1,2]**, Frédéric Barras**[2]**, Simonetta Gribaldo**[1] *

**1** Institut Pasteur, Université Paris Cité, Department of Microbiology, Unit Evolutionary Biology of the Microbial Cell, Paris, France, **2** Institut Pasteur, Université Paris Cité, CNRS UMR6047, Department of Microbiology, Unit Stress Adaptation and Metabolism in enterobacteria, Paris, France

* simonetta.gribaldo@pasteur.fr

**Data Availability Statement:** All relevant data are within the paper and its Supporting Information files.

## Abstract

Establishing the origin of mitochondria and plastids is key to understand 2 founding events in the origin and early evolution of eukaryotes. Recent advances in the exploration of microbial diversity and in phylogenomics approaches have indicated a deep origin of mitochondria and plastids during the diversification of *Alphaproteobacteria* and *Cyanobacteria*, respectively. Here, we strongly support these placements by analyzing the machineries for assembly of iron–sulfur ([Fe–S]) clusters, an essential function in eukaryotic cells that is carried out in mitochondria by the ISC machinery and in plastids by the SUF machinery. We assessed the taxonomic distribution of ISC and SUF in representatives of major eukaryotic supergroups and analyzed the phylogenetic relationships with their prokaryotic homologues. Concatenation datasets of core ISC proteins show an early branching of mitochondria within *Alphaproteobacteria*, right after the emergence of *Magnetococcales*. Similar analyses with the SUF machinery place primary plastids as sister to *Gloeomargarita* within *Cyanobacteria*. Our results add to the growing evidence of an early emergence of primary organelles and show that the analysis of essential machineries of endosymbiotic origin provide a robust signal to resolve ancient and fundamental steps in eukaryotic evolution.

Endosymbiosis events were important steps in the origin and early evolution of eukaryotes and notably those that gave rise to mitochondria and primary plastids from *Alphaproteobacteria* and *Cyanobacteria*, respectively [1–3]. Recent phylogenomic analyses have suggested a basal placement of mitochondria with *Alphaproteobacteria* and that the long-time assumed branching with the *Rickettsiales* is likely due to a compositional bias driven by convergent AT-rich genomes [4–8]. Concerning the placement of plastids, several analyses have supported the basal position of plastids [9,10] and, more recently, as sister to the newly sequenced cyanobacterium *Gloeomargarita lithophora* [11–13].

Iron–sulfur [Fe–S] clusters are very ancient protein cofactors essential for life and are involved in many biological processes [14,15]. Our recent analyses have reconstructed the

**Funding:** This work was supported by the European Union's Horizon 2020 research and innovation program under grant 722361 (salary of PSG) and the Agence Nationale de la Recherche ANR-10-LABX- 62-IBEID to FB and SG. The funders had no role in study design, data collection and analysis, decision to publish, or preparation of the manuscript.

**Competing interests:** The authors have declared that no competing interests exist.

taxonomic distribution and evolution of the characterized machineries, SUF and ISC, in Bacteria [16]. In eukaryotes, SUF and ISC operate in plastids and mitochondria, respectively [17], and catalyze similar biochemical steps but involve different proteins (Figs 1A and 2A). The components of the ISC machinery are encoded by nuclear genes and are translocated to mitochondria to synthesize [Fe–S] clusters [18]. The SUF components are also mostly encoded by nuclear genes, to the exception of SufB and SufC, which can be encoded in plastid genomes [19,20]. From previous phylogenetic analyses of some components, it is generally accepted that these machineries originated from the 2 endosymbionts [17,21–24]. However, the phylogenetic signal brought by these systems as a whole has not been used to investigate the placement of mitochondria and plastids.

We used our recent datasets [16] to analyze in detail the taxonomic distribution of the ISC and SUF machineries in 1,191 genomes covering the current diversity of *Alphaproteobacteria*

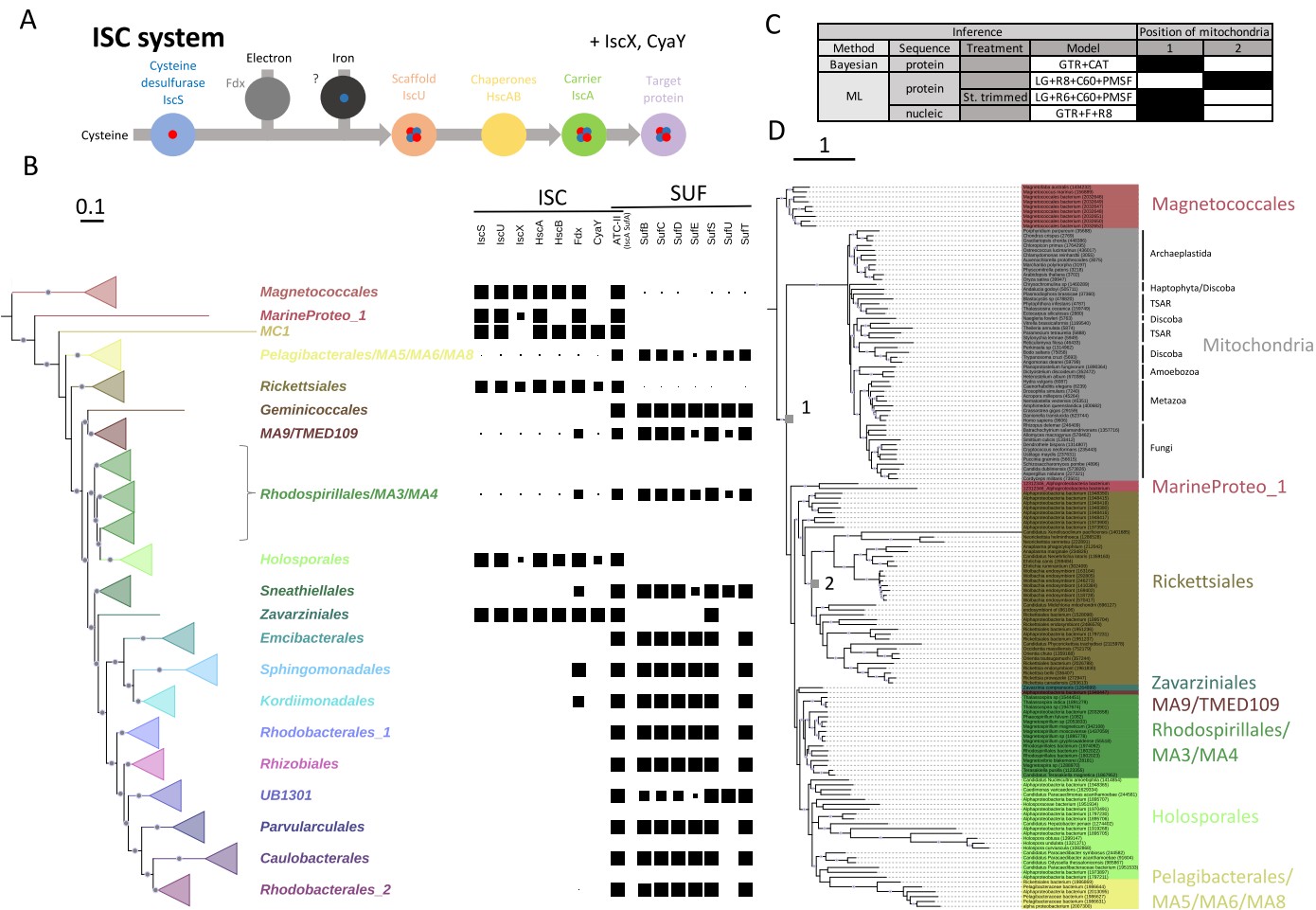

**Fig 1. (A)** Schematic view of [Fe–S] cluster biosynthesis by the ISC system and the corresponding components. **(B)** Taxonomic distribution of the ISC and SUF systems mapped on the *Alphaproteobacteria* reference tree (IQ-TREE, LG+R10+C60+PMSF. IF2+RpoB+RpoC, 3,429 amino acid positions, 1,193 sequences). Dots at branches indicate ultrafast-bootstrap values ≥ 95%. The scale bar indicates average number of substitutions per site. **(C)** Summary of the placement of mitochondria using different approaches and models. Two alternative positions are indicated and reported in (D). **(D)** Bayesian phylogeny from the concatenated ISC dataset including alphaproteobacterial and eukaryote homologues (IscA+IscS+HscA+Fdx+CyaY, Phylobayes, GTR+CAT, 1,306 amino acid positions, 149 sequences). Dots at branches indicate posterior probabilities ≥ 0.95. The scale bar indicates average number of substitutions per site. Numbers at the tips indicate the taxonomy IDs from NCBI. The data underlying this Figure can be found in S1 Data.

(for a list of taxa, see S1 Table). The presence of ISC and SUF homologues in *Alphaproteobacteria* is patchy, ISC being identified mostly in the basal lineages (*Magnetococcales*, Marine *Proteobacteria* 1, MC1) and in *Rickettsiales*, whereas it is absent in most of the other *Alphaproteobacteria* that have the SUF system (Fig 1B and S2 Table). The pattern of mutual exclusion between ISC and SUF has been observed in other bacterial groups, especially *Gammaproteobacteria* [16]. A phylogeny based on concatenation of 5 core ISC proteins (IscS, HscA, Fdx, CyaY, and IscA, 1,306 amino acid positions) is largely consistent with the reference *Alphaproteobacteria* phylogeny, showing the monophyly of all major orders (S1 Fig). These results indicate that the ISC system was present in the ancestor of *Alphaproteobacteria* and was inherited mainly vertically, while it was subsequently lost in many lineages during diversification of this phylum. This distribution de facto excludes an origin of the mitochondrial ISC from many lineages.

We then investigated the presence of the SUF and ISC systems in 66 genomes covering the main eukaryotic phyla [25] (for a list of taxa, see S3 Table). Homologues of the 8 ISC machinery components were identified in most eukaryotic taxa—except for IscX—and are encoded in nuclear genomes (S4 Table). Preliminary single-gene tree analyses allowed to clearly identify the eukaryotic orthologues of mitochondrial origin by their branching with *Alphaproteobacteria* (S1 Data). We therefore added these ISC components in the *Alphaproteobacteria* dataset to investigate their placement. Bayesian analysis using the CAT+GTR model supports the monophyly of eukaryotic sequences (PP = 1) and their deep branching, just after *Magnetococcales* (PP = 0.82). These results strengthen the *Alphaproteobacteria*-deep hypothesis, although the position of eukaryotes is more basal than previously observed [4,6], branching before the Marine Proteobacteria 1 group (Fig 1C and 1D). Moreover, the internal topology of *Alphaproteobacteria* agrees with the reference phylogeny of this phylum, notably with *Rickettsiales* branching after *Magnetococcales* and sister to all remaining *Alphaproteobacteria* (Fig 1D).

We investigated the robustness of the *Alphaproteobacteria*-deep placement by analyzing the ISC dataset with a panel of alternative models and methodologies (Fig 1C). An ML phylogeny with the LG+R8+C60+PMSF model indeed shows Eukaryotes as sister group of *Rickettsiales*, although with low support (bootstrap value (BV) = 48%) (Fig 1C, position #2 in Fig 1D, S2 Fig). Moreover, the internal phylogeny of *Alphaproteobacteria* shows incongruencies with both the reference phylogeny of the phylum and the alphaproteobacterial ISC concatenation tree, notably with the split of the *Rickettsiales* in 2 groups (S2 Fig), which strongly suggest the presence of a tree reconstruction artefact using this model when eukaryotic sequences are included.

Given that ISC components are all nuclear encoded, the "*Rickettsiales*-sister" placement is unlikely to be due to a convergent compositional bias toward AT-rich genomes between *Rickettsiales* and mitochondria. An AminoGC plot (S3A Fig) shows that the GC bias of eukaryotic sequences is not particularly similar neither to *Rickettsiales* or *Magnetococcales*. The removal of compositional heterogeneous sites by a stationary-based method resulted in a tree consistent with the Bayesian tree, placing again Eukaryotes at the base of *Alphaproteobacteria*, and after the *Magnetococcales*, although with low support (BV = 23%), but shows a correct internal phylogeny of *Alphaproteobacteria* (Figs 1C and S4). An AminoGC plot of the trimmed dataset shows a reduced difference between the groups, suggesting that lowering the GC bias between clades might remove the incorrect placement of Eukaryotes with *Rickettsiales* (S3B Fig). Surprisingly, despite the fact that nucleic sequences are usually more prompt to compositional bias, an ML tree obtained from the nucleic acid version of the original alignment (GTR model) also strongly supports the "*Alphaproteobacteria*-deep" placement (BV = 100%), although with a poorly resolved internal topology of *Alphaproteobacteria* (S5 Fig). A %GC plot of this dataset shows a similar pattern as the full protein dataset (S3C Fig).

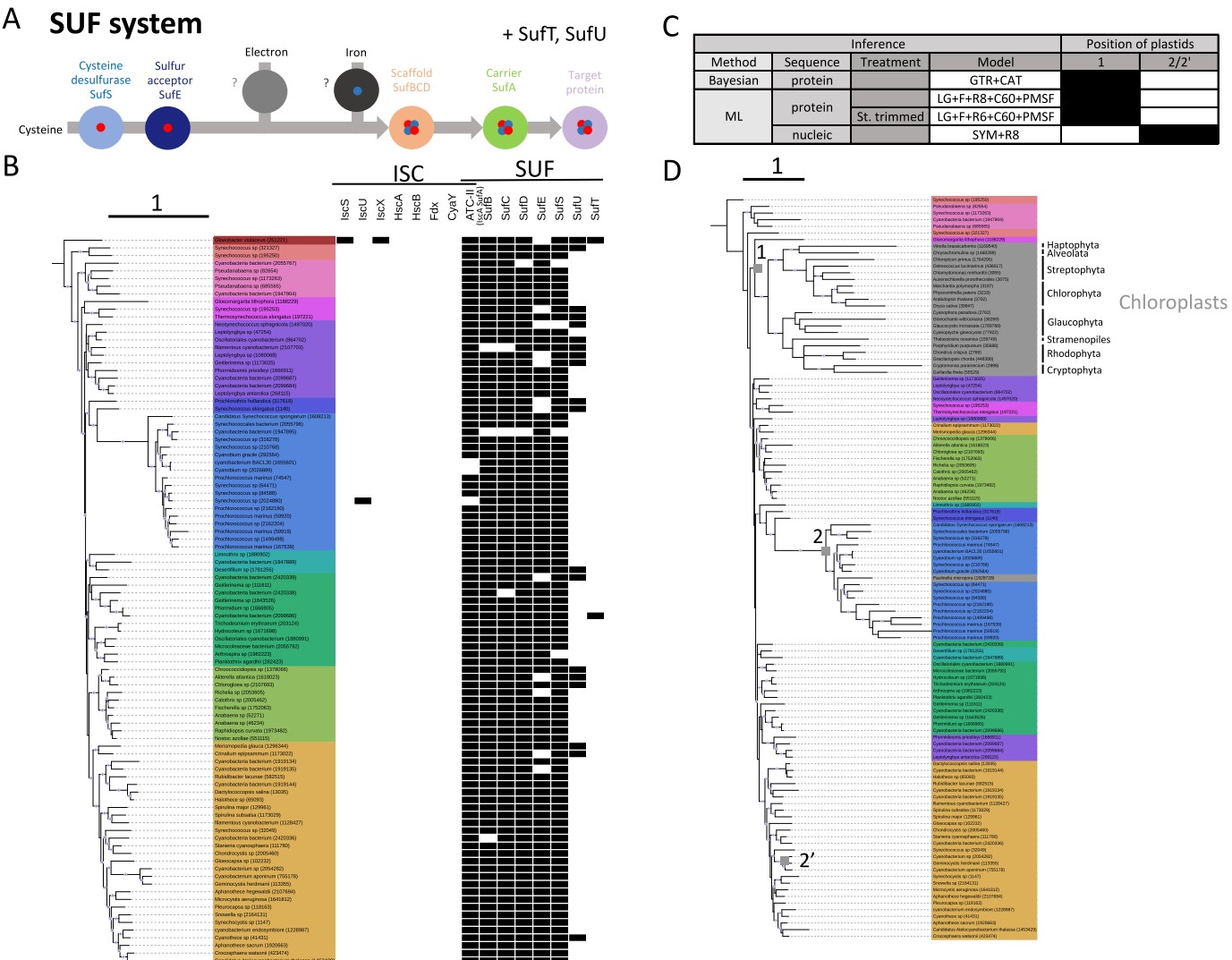

**Fig 2. (A)** Schematic view of [Fe–S] cluster biosynthesis by the SUF system and the corresponding components. **(B)** Taxonomic distribution of the ISC and SUF systems mapped on the *Cyanobacteria* reference tree (IQ-TREE, LG+R10+C60+PMSF. IF2+RpoB+RpoC. 3,026 amino acid positions, 107 sequences). Dots at the branches indicate ultrafast-bootstrap values ≥ 95%. The scale bar indicates average number of substitutions per site. **(C)** Summary of the placement of mitochondria using different approaches and models. Three alternative positions are indicated and reported in (D). **(D)** Bayesian phylogeny from the concatenated SUF dataset including cyanobacterial and eukaryote homologues (SufB+SufC+SufD+SufS) (Phylobayes, GTR+CAT, 1,565 amino acid positions, 112 sequences). Dots at branches indicate posterior probabilities ≥ 0.95. The scale bar indicates average number of substitutions per site. Numbers at the tips indicate the taxonomy IDs from NCBI. The data underlying this Figure can be found in S1 Data.

Altogether, these results suggest that sequence compositional bias is at least partially responsible for the placement of Eukaryotes with the *Rickettsiales*, in agreement with recent analyses [4,6,7]. This tree reconstruction artefact can be counterbalanced either by removing compositional heterogeneous positions or by using the GTR model, whereas ML site heterogeneous models such as LG+C60+PMSF fail to tackle this issue.

We then used the same approach to investigate the origin of plastids by analyzing the SUF system (Fig 2). We used the dataset from our previous study [16], originating from 95 genomes covering the current diversity of *Cyanobacteria* (for a list of taxa, see S5 Table). Whereas the SUF system is largely present in most genomes, the ISC system is absent (Fig 2B and S6 Table).

An ML tree obtained from concatenation of the 4 most conserved SUF markers (SufB, SufC, SufD, and SufS, 1,561 amino acid positions) is largely consistent with the reference *Cyanobacteria* phylogeny, except a few misplacements likely due to specific intra-phylum HGTs (S6 Fig). We did not include *Gloeobacter* in the dataset as we have recently shown that its original SUF was replaced by one laterally acquired from other bacteria [16].

Among Eukaryotes, SUF homologues are present only in photosynthetic lineages, either nuclear or plastid encoded (S4 Table). Preliminary single-gene tree analyses allowed us to identify the eukaryotic orthologues of plastid origin by their branching with *Cyanobacteria* (S1 Data). When included in the cyanobacterial dataset, Bayesian analysis (CAT+GTR) (Fig 2D) and ML analysis (LG+C60+PMSF) without and with stationary-based trimming (S7 and S8 Figs, respectively) support a deep placement of plastids within *Cyanobacteria* (position #1 in Fig 2C and 2D), in agreement with previous analyses [9,10]. Interestingly, these trees favor the placement of plastids as sister to *Gloeomargarita lithophora* (PP = 0.65, BV = 70%, BV = 54%), strengthening recent data [11–13]. Puzzlingly, ML analysis with nucleic acids (S9 Fig) infers a highly incongruent tree where Eukaryotes are not monophyletic and branch with different cyanobacterial clades (positions #2 and #2' in Fig 2C and 2D), which strongly indicates that this tree is not reliable. Finally, our data nicely confirm the independent acquisition of a primary plastid in the amoeba *Paulinella microspora* from a member of *Prochlorococcales* [26,27] (Figs 2D and S7–S9).

The origins of mitochondria and plastids is a difficult question to address phylogenetically, due to the antiquity of such events and the potential biases in composition and evolutionary patterns arising from profound adaptations that occurred during the endosymbiosis process. Recent advances in genomic coverage from *Alphaproteobacteria* and *Cyanobacteria*, together with improvement of phylogenomics approaches and evolutionary models, have allowed to clarify the timing and origin of these endosymbioses. We show here that the [Fe–S] cluster biosynthesis ISC and SUF machineries provide a robust additional dataset to infer the origin of organelles. Four criteria concur to the pertinence of using these machineries: (i) they were inherited from *Alphaproteobacteria* and *Cyanobacteria* at the origin of Eukaryotes and primary photosynthetic lineages, respectively; (ii) they carry out an essential cellular function; (iii) most of their components are encoded in the nucleus, reducing the problem of compositional biases; and (iv) being part of a highly integrated process, they were likely subjected to similar evolutionary constraints.

The complexity of the large protein families, including the ISC and SUF components, may have prevented their selection in previous large-scale automated analyses searching for organelle orthologues. Therefore, a similar approach focusing on the detailed analysis of other fundamental eukaryotic systems of mitochondrial and plastid origin, coupled with increase in genomic coverage from deep branches of the *Alphaproteobacteria* and *Cyanobacteria*, will surely provide further key information on our most ancient past.

## Material and methods

### Genome databases

We used a database of 10,865 archaeal and bacterial proteomes, which we recently assembled for our survey of [Fe–S] biosynthesis machineries in prokaryotes [16]. The database contains 1,188 proteomes annotated as *Alphaproteobacteria* (S1 Table) and 95 proteomes annotated as *Cyanobacteria* (S5 Table). For the present analysis, we added 3 alphaproteobacterial Metagenome Assembled Genomes from the Martijn study (one corresponding to MarineAlpha4 and two to MarineProteo1) [4]. We also assembled a database of 66 Eukaryote proteomes gathered from the NCBI and representing the major eukaryotic superphyla (TSAR, *Amorphea*,

*Excavates*, *Cryptista*, *Haptista*, *Archaeplastida*) [25] (S3 Table). We also included 4 plastid genomes from Glaucophyta (*Cyanophora paradoxa*, *Gloeochaete wittrockiana*, *Glaucocystis incrassata*, *Cyanoptyche gloeocystis*).

## Assembly of datasets

Datasets of prokaryotic homologues of IscA, IscS, IscU, IscX, CyaY, HscA, HscB, Fdx, SufB, SufC, SufD, SufE, SufS, SufT, SufU were already assembled as described in [16]. Here, for eukaryote sequences, we used the same procedure. Briefly, we used HMM profiles of each component from [16] to perform an HMM search using HMMER v3.2.1 [28] on the eukaryotic database, selecting hits with an *e*-value < 0.01. These hits were then added to the prokaryotic homologue datasets. Sequences were aligned using MAFFT v7.419 [29] (auto option), the alignments were manually curated to eliminate nonhomologous sequences, and preliminary phylogenies were inferred using FASTTREE v2.1.10 [30] (LG+G4) and with and without trimming using BMGE v1.12 [31] (BLOSUM30). For each component, eukaryotic orthologue subfamilies were delineated manually based on the branching of sequences within *Cyanobacteria* or *Alphaproteobacteria*, taxonomic distribution, domain composition, and length of sequences. We did not find any homologs of the archaeal SMS system [16], to the exception of SmsB and SmsC, which are fused in the same ORF in the genome of *Blastocystis* sp. ATCC 50177, as previously reported [32]. The SufB and SufC of the 4 Glaucophyta plastid genomes were identified using tBLASTn [33] and added manually. All preliminary trees used for the delineation of eukaryote orthologue groups are available in S1 Data.

For the concatenation, we selected the markers based on different criteria. Markers that were not broadly distributed (IscX in eukaryotes, SufU and SufT in *Cyanobacteria*) were eliminated. We also discarded HscB and IscU as they did not form clear monophyletic groups in preliminary trees. Finally, in eukaryotes, 2 homologues of IscA (ISA1 and ISA2 belonging to the large ATC-II and ATC-I protein subfamilies, respectively [22]), SufE and SufS, were identified. For IscA (ISA) and SufS, we selected the paralogues distributed similarly to other ISC/SUF components and which branch with *Alphaproteobacteria* and *Cyanobacteria*, respectively, in the preliminary phylogenies (ISA1 and SufS1 in S4 Table). We discarded the whole SufE family, as it included 2 clades (SufE1 and SufE2 in S4 Table) either containing multiple paralogues or not widely conserved in Eukaryotes. Although the ATC-II family is shared by the ISC and SUF systems (IscA and SufA) in *Alphaproteobacteria*, we selected IscA for the ISC concatenation as we observed that it is in vicinity of the rest of ISC system [16] and follows the reference tree of *Alphaproteobacteria* group.

Finally, alignments and each protein family dataset were aligned using MAFFT v7.419 [29] (LINSI option) and trimmed using BMGE v1.12 [31] (entropy threshold = 0.95, minimum length = 1, matrix = BLOSUM30) and individual duplicated sequences (paralogues, isoforms, assembly artifacts) were removed after visual inspection of trees/alignments. The nucleic sequences were back-aligned on amino acid sequences by converting each amino acid in its respective codon by a custom script (S2 Data). For trimming of the nucleic alignments, we used the -t CODON option of BMGE. For the concatenations, we kept a taxon if it possessed n (markers) ≥ 3 for both ISC and SUF, except for Rhodophyta and Glaucophyta (SUF) for which we retained the only 2 detected markers. The highly divergent sequences from amitochondriate eukaryotes (*Metanomada* and *Entamoeba*) were removed to avoid tree reconstruction artefacts.

For the reference trees of *Alphaproteobacteria* and *Cyanobacteria*, we assembled supermatrices using IF2, RpoB, and RpoC as markers with the same procedure as described above.

## Phylogenetic inference

The ML phylogenies of ISC and SUF systems based on protein sequences were inferred using IQ-TREE v1.6.10 [34], with the best model according to BIC criteria and with the PMSF method (posterior mean site frequency) option with 60 mixture categories [35], with the starting phylogenies inferred by homogeneous models. To assess the robustness of branches, 100 nonparametric bootstrap replicates were used. The tree of SUF was rooted using *Nitrosomonadales* and *Balneolaeota*, as SUF was anciently acquired by horizontal gene transfer from these organisms [16] (S7 Table). SUF homologues from *Gloeobacter* were removed as the original system was replaced in this bacterium by HGT from other bacteria [16]. The ML phylogenies based on nucleic sequences were inferred using IQ-TREE, with the GTR/SYM models. The Bayesian phylogenies were inferred using Phylobayes v4.1c [36], with the GTR+CAT model. For ISC, 4 chains were run for around 99,000 iterations each. The convergence between chains was tested using bpdiff with a sampling of 1,885 and 1,878 trees (every 50 trees) and a burn-in of 5,000. Two chains with a maxdiff < 0.15 (0.13) were used to infer the consensus tree. The other maxdiff results correspond to lower but acceptable convergence (0.23, 0.25, 0.22, 0.32, 0.32). For SUF, 4 chains were run for around 86,000 iterations each. The convergence between chains was tested using bpdiff with a sampling of 14,259 and 14,235 trees (every 5 trees) and a burn-in of 15,000. Two chains with a maxdiff ≈ 0.3 (0.3079) were used to infer the consensus tree. The rest of the other maxdiff results correspond to nonconvergent runs (0.75, 0.45, 0.87, 0.58, 0.60).

The ML reference phylogenies of *Alphaproteobacteria* and *Cyanobacteria* were inferred using IQ-TREE v1.6.10 [34], with the best model according to BIC criteria and with the PMSF method (posterior mean site frequency) option with 60 mixture categories [35] with the starting phylogenies inferred by homogeneous models. To assess the robustness of branches, 1,000 fast-bootstrap replicates were used. The two reference trees were rooted using as outgroup other *Proteobacteria* and *Melainabacteria*, respectively (S7 Table).

## Supporting information

**S1 Fig. (A)** Reference tree of *Alphaproteobacteria* possessing an ISC system (IF2+RpoB +RpoC). IQ-TREE, LG+F+R7+C60+PMSF, 3,483 amino acid positions, 96 sequences. Numbers at branches indicate ultrafast-bootstrap values. The scale bar indicates average number of substitutions per site. Numbers at the tips indicate taxonomy IDs from NCBI. **(B)** ML Phylogeny of alphaproteobacterial ISC (IscA+IscS+HscA+Fdx+CyaY). IQ-TREE, LG+R7+C60 +PMSF, 1,275 amino acid positions, 97 sequences. Numbers at branches indicate nonparametric bootstrap values. The scale bar indicates average number of substitutions per site. Numbers at the tips indicate taxonomy IDs from NCBI. The data underlying this Figure can be found in S1 Data.
(TIFF)

**S2 Fig. ML phylogeny of alphaproteobacterial and eukaryote ISC (IscA+IscS+HscA+Fdx +CyaY).** IQ-TREE, LG+R8+C60+PMSF, 1,306 amino acid positions, 149 sequences. Numbers at branches indicate nonparametric bootstrap values. The scale bar indicates average number of substitutions per site. Numbers at the tips indicate taxonomy IDs from NCBI. The data underlying this Figure can be found in S1 Data.
(TIFF)

**S3 Fig. Boxplot of AminoGC and %GC distribution among sequences for the ISC concatenation. (A)** AminoGC for protein concatenation. **(B)** AminoGC for protein concatenation after stationary-based trimming. **(C)** %GC for nucleic protein concatenation. The AminoGC

was calculated by the script prune_ali.pl from Martijn and colleagues [4]. Middle bar: median; hinges: first and third quartiles; whiskers: largest value no further than 1.5 (interquartile range); dots: individual points; big dots: outliers. The upper bars correspond to the two-tailed Mann–Whitney U test result ($H_0$ = no difference between the 2 groups). N.S.: $p > 0.05$, *: $p < 1 \times 10^{-3}$, **: $p < 1 \times 10^{-5}$.
(TIFF)

**S4 Fig. ML phylogeny of alphaproteobacterial and eukaryote ISC with stationary-based trimming.** IQ-TREE, LG+R6+C60+PMSF, 532 amino acid positions, 149 sequences. Numbers at branches indicate the nonparametric bootstrap values. The scale bar indicates average number of substitutions per site. Numbers at the tips indicate taxonomy IDs from NCBI. The data underlying this Figure can be found in S1 Data.
(TIFF)

**S5 Fig. ML nucleic acid phylogeny of alphaproteobacterial and eukaryote ISC (IscA+IscS +HscA+Fdx+CyaY).** IQ-TREE, GTR+F+R8, 4,128 nucleic positions, 149 sequences. Numbers at the branches indicate nonparametric bootstrap values. The scale bar indicates average number of substitutions per site. Numbers at the tips indicate taxonomy IDs from NCBI. The data underlying this Figure can be found in S1 Data.
(TIFF)

**S6 Fig. ML phylogeny of cyanobacterial SUF rooted with an external group (SufB+SufC +SufD+SufS).** IQ-TREE, LG+F+R10+C60+PMSF. 1,561 amino acid positions, 167 sequences. The numbers at the branches indicate the ultrafast-bootstrap values. The scale bar indicates the average number of substitutions per site. Numbers at the tips indicate taxonomy IDs from NCBI. The data underlying this Figure can be found in S1 Data.
(TIFF)

**S7 Fig. ML phylogenies of cyanobacterial and eukaryote SUF (SufB+SufC+SufD+SufS).** **(A)** ML phylogeny of cyanobacterial and eukaryote SUF rooted with an external group (without stationary-based trimming). IQ-TREE, LG+R10+C60+PMSF. 1,553 amino acid positions, 188 sequences. The numbers at the branches indicate the nonparametric bootstrap values. The scale bar indicates the average number of substitutions per site. Numbers at the tips indicate taxonomy IDs from NCBI. **(B)** ML phylogeny of cyanobacterial and eukaryote SUF (without stationary-based trimming). IQ-TREE, LG+R8+C60+PMSF. SufB+SufC+SufD+SufS. 1,565 amino acid positions, 112 sequences. The numbers at the branches indicate the nonparametric bootstrap values. The scale bar indicates the average number of substitutions per site. Numbers at the tips indicate taxonomy IDs from NCBI. The data underlying this Figure can be found in S1 Data.
(TIFF)

**S8 Fig. ML phylogeny of cyanobacterial and eukaryote SUF (with stationary-based trimming).** IQ-TREE, LG+R6+C60+PMSF. SufB+SufC+SufD+SufS. 813 amino acid positions, 112 sequences. The numbers at the branches indicate the nonparametric bootstrap values. The scale bar indicates the average number of substitutions per site. Numbers at the tips indicate taxonomy IDs from NCBI. The data underlying this Figure can be found in S1 Data.
(TIFF)

**S9 Fig. ML nucleic phylogeny of cyanobacterial and eukaryote SUF.** IQ-TREE, SYM+R8. SufB+SufC+SufD+SufS. 4,809 nucleic positions, 112 sequences. The numbers at the branches indicate the nonparametric bootstrap values. The scale bar indicates the average number of substitutions per site. Numbers at the tips indicate taxonomy IDs from NCBI. The data

underlying this Figure can be found in S1 Data.
(TIFF)

**S1 Table. List of alphaproteobacterial proteomes used in this study.**
(XLSX)

**S2 Table. Complete taxonomic distribution of the different proteins (ISC and SUF system) in *Alphaproteobacteria*.**
(XLSX)

**S3 Table. List of eukaryote proteomes and genomes used in this study.**
(XLSX)

**S4 Table. Complete taxonomic distribution of the different proteins (ISC and SUF system) in Eukaryotes.**
(XLSX)

**S5 Table. List of cyanobacterial proteomes used in this study.**
(XLSX)

**S6 Table. Complete taxonomic distribution of the different proteins (ISC and SUF system) in *Cyanobacteria*.**
(XLSX)

**S7 Table. List of taxa used as outgroups.**
(XLSX)

**S1 Data. Sequences, alignments, and phylogenies used in this study.**
(ZIP)

**S2 Data. Custom script used in this study.**
(ZIP)

## Author Contributions

**Conceptualization:** Pierre Simon Garcia, Simonetta Gribaldo.

**Formal analysis:** Pierre Simon Garcia.

**Methodology:** Pierre Simon Garcia.

**Supervision:** Simonetta Gribaldo.

**Validation:** Pierre Simon Garcia, Simonetta Gribaldo.

**Writing – original draft:** Pierre Simon Garcia, Simonetta Gribaldo.

**Writing – review & editing:** Pierre Simon Garcia, Frédéric Barras, Simonetta Gribaldo.

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
