## [Editor Report · Decision Letter 0]

3 Mar 2023

Dear Simonetta, 

Thank you for submitting your manuscript entitled "Iron-Sulfur cluster biosynthesis machineries support a deep origin of mitochondria and plastids" for consideration as a Short Reports by PLOS Biology.

Your manuscript has now been evaluated by the PLOS Biology editorial staff, and I'm writing to let you know that we would like to send your submission out for external peer review.

Once your full submission is complete, your paper will undergo a series of checks in preparation for peer review. After your manuscript has passed the checks it will be sent out for review. To provide the metadata for your submission, please Login to Editorial Manager (https://www.editorialmanager.com/pbiology) within two working days, i.e. by Mar 07 2023 11:59PM.

Kind regards,

Roli

Roland Roberts, PhD

Senior Editor

PLOS Biology

rroberts@plos.org

---

## [Decision Letter · Decision Letter 1]

10 May 2023

Dear Simonetta,

Thank you for your patience while your manuscript "Iron-Sulfur cluster biosynthesis machineries support a deep origin of mitochondria and plastids" was peer-reviewed at PLOS Biology. It has now been evaluated by the PLOS Biology editors, an Academic Editor with relevant expertise, and by three independent reviewers. Please accept my apologies for the delay incurred while we sought additional advice regarding the decision.

In light of the reviews, which you will find at the end of this email, we would like to invite you to revise the work to thoroughly address the reviewers' reports.

You’ll see that reviewer #1 is broadly positive, but wants you to add glaucophytes to the plastid analyses, and indeed found your taxon selection a little disappointing; they also request more discussion of repeated replacement of ISC by DUF, and want you to do more appropriate bootstrapping. Reviewer #2 is also favourable, and has only minor requests. Reviewer #3, however, is strongly negative, raising significant concerns about your interpretation of the results, and your claims of novelty with respect to the existing literature. This reviewer also suggests that you show a tendency to dismiss other people’s markers with poor or ad hoc arguments while presenting weak or flawed arguments for your own, downplaying inconsistencies among your own results. Clearly there is a significant divergence in views here, but after extensive discussion, we have decided to give you an opportunity to address and/or rebut the severe concerns raised by reviewer #3.

Given the extent of revision needed, we cannot make a decision about publication until we have seen the revised manuscript and your response to the reviewers' comments. Your revised manuscript is likely to be sent for further evaluation by all or a subset of the reviewers.

**IMPORTANT - SUBMITTING YOUR REVISION**

*Re-submission Checklist*

*Published Peer Review*

*PLOS Data Policy*

*Blot and Gel Data Policy*

Sincerely,

Roli

Roland Roberts, PhD

Senior Editor

PLOS Biology

rroberts@plos.org

REVIEWERS' COMMENTS:

Reviewer #1:

The authors present an analysis of iron-sulfur cluster biosynthesis genes to look at the endosymbiotic origin of plastids and mitochondria. The evolution of these organelles has been controversial pretty much forever - their endosymbiotic origin is now beyond any serious question, but this itself has raised new questions and they are also sometimes tied to debates about the origin of eukaryotes as a whole. Any new insights on this subject are potentially important and significant, so evolutionary biologists have been attacking this from several directions, mostly using genomics. One approach of late has been to use nucleus-encoded genes for proteins targeted to the organelles to assess their origins, since organelle genomes may be more direct, but their sequence evolution has been odd and confounding. Here, the authors use a pathway share by both organelles, but using different non-homologous versions of the pathway, to see if it sheds any light on their origin. This does not actually touch on some of the more hotly debated topics (e.g. the whole "mitochondrion-early vs -late debate that seems to never end), but it is useful as the results seems relatively solid and they do touch on other debates such as the nearest bacterial relatives of these organelles. The results here are not surprising in that they seem to support existing hypotheses, but I think this should not be a strike against the work as we do tend to favour "surprising" and likely wrong things over actually supporting something that is less surprising but real in this field. The main results of interest to my mind are the position of the mitochondria deep and not with the rickettsiales (the traditional placement, but one that was always potentially a long-branch or compositional bias artifact). This is not the first paper to suggest this, but this is a very big shift should this come to be the favoured explanation of the data, so it is quite important. The paper is short, and this is a suitable length since they authors explain what needs explaining for the most part. I have a few suggestions however, some of which are pretty important I think. These are in no particular order, so I will try to note the most important. 

1. I think the main problem i have is the lack of glaucophytes in the plastid analysis. These are one of only three primary plastid groups, and the order of those three is hotly contested. Why are they absent? Is there a reason, if so I missed it. If not, I am sorry to say I think they really need to be added and their relationship with the reds and greens noted. This might even prove to be an important point depending on what the analyses say. Also at least need to see if they change anything else. Overall the taxon selection is pretty low considering all the data around these days (from the plastid-containing eukaryotes, the taxon selection in the bacteria is much better). 

2. The SUF/ISC distribution in alphaproteobacteria is interesting. The contention here is clearly that ISC is ancestral and was replaced repeatedly by SUF. Is this repeated replacement true in other groups? Is this something that is already observed and tested? It seems like an important point that needs slightly more discussion and references if it is already discussed elsewhere. 

3. The point that "although another study has suggested that they branch with a more distal group11" needs to be more specific. Please spell out where they went in this analysis.

4. In the figures (part D of both), what are the numbers on the taxon names? It looks kind of "raw". I suggest the authors should take a bit more time to clean up the figure to make it easier for the non-expert to read. Get rid of internal numbering and underscores etc that mean nothing to the reader. The figure looks like "direct from analysis", and not a finished figure. I always think the reader deserves a bit of work to make the data easy to grasp. 

5. (Unfortunately the pages are not numbered so I am going from the PDF off the web site) On page 13 and 14 the authors note they did "1,000 fast-bootstrap replicates" with IQ-tree. these are something specific to this program and need to be interpreted differently from "normal" bootstraps so this should be noted (they are inflated). You can run non-parametric bootstraps with PMSF, and they are not as inflated, so that is worth considering. 

6. On page 14 the Bayesian analyses are described based on two chains, what happened with the other chains? Did they all converge? It is important if they don't to say what they did. 

Reviewer #2:

The paper by Garcia et al deals with the origin of the Iron-sulfur cluster machineries at the onset of mitochondria and plastids. In this paper, the authors extended their previous prokaryotic analysis regarding the different iron sulfur clusters biosynthesis pathways to eukaryotes and, based on a careful phylogenetic analysis of the core ISC and SUF components, have retrieved a deep emergence of mitochondria after the divergence of Magnetococcales and plastids, as a sister clade of Cyanobacteria.

As the authors point out, in the last years there has been several papers addressing the origin of mitochondria (and plastids) with contradictory results. We should keep in mind that phylogenetic analysis are a proxy for likelihoods and not certainties. Moreover, they are highly dependent on the methods and datasets used. I doubt that this paper will solve the current literature debate, but will surely contribute to the discussion. 

Having said that, this is a timely paper, that uses nuclear encoded genes as proxy for the origin of mitochondria and plastids. The analyses presented in here are careful and all data to retrieve the authors steps is provided in supplementary information. As also observed by the authors, different methods, can give raise to different results. 

I have only some suggestions with the aim to clarify and further support the authors methodology. 

- It would be relevant, to present as supplementary figures and/or in newick/figtree format, the preliminary single gene phylogenies containing also the paralogous that were excluded, as well as the non-monophyletic behaviour of some of the proteins (IscU, HscB). This will give strength to the methodology used. 

- An UFB support of 35% is meaningless. According to the IQTree authors, UFB below 95% are not significant (at least for the current version). Even if in the sentence regarding the stationary-based method trimming ML phylogeny, reference to the poor support is given, this could perhaps be rephrased to indicate that in this phylogeny, the branching of eukaryotes is not clear (and not start the sentence with In contrast…)

Reviewer #3:

In this work, Garcia and colleagues attempt to use two sets of proteins belong to the iron-sulfur cluster machineries ISC and SUF, to infer the phylogenetic origin of the mitochondrion and plastid, respectively.

While the methods are sound, there are profound issues with the interpretation of the results and their novelty in comparison to what is already known. 

--- Misrepresentation of the state-of-the-art ---

The authors exaggerate the real existence of a debate when it comes to the origin of mitochondria and plastids. While mitochondrial origins WERE debated for a long time, the two most sophisticated analyses on these questions in terms of complex phylogenetic approaches and broad taxon sampling (Martjin, Nature 2018; Munoz-Gomez NEE 2022) both agree on the deep phylogenetic placement of mitochondria and have given strong methodological arguments for the invalidity of the results presented elsewhere.

When it comes to plastids, since the discovery of Gloeomargarita (Ponce-Toledo 2016), there has been NO conflicting reports about their status as the closest relatives to plastids. Their relationship to plastids is even easily recovered and supported in 16S+23S phylogenies. Any "debate" that might have existed has been put to rest since 2016.

In addition, the writing about what is known on the origin of the SUF and ISC eukaryotic machineries is somewhat unclear. It has long been reported that they are of plastid and mitochondrial origin, respectively. However, when the authors report their results here, they fail to make it clear that there was no doubt about this. For example "Among eukaryotes, SUF homologues are present only in photosynthetic lineages, and preliminary single gene tree analyses place them with Cyanobacteria, confirming their plastid origin." This has long been known, and reported in one of the reviews cited here and references therein (Tsaousis 2019).

Even the distribution of the ISC and SUF proteins in Alphaproteo and Cyanobacteria has been reported elsewhere by the same authors (Garcia NEE 2022) and yet here, the authors present it as if it were new. E.g. "We then used the same approach to investigate the origin of plastids by analyzing the SUF system. 95 genomes covering the current diversity of Cyanobacteria were scrutinized. Whereas the SUF system is largely present in most genomes, the ISC system is practically absent."

--- Ad hoc arguments against previously used phylogenetic markers ---

The authors state that "These previous studies relied on the use of large concatenations of protein markers involved in various cellular processes and encoded either in the nucleus or the organelle, which likely drives different evolutionary constraints that may lead to phylogenetic instability." without providing further proof that this is the case and simply use this as a reason for why there is a need for other phylogenetic markers (i.e. theirs).

--- Arguments for the using the SUF and ISC proteins as "better phylogenetic markers" are flawed ---

At the end of this paper, the authors summarize the four reasons for using the SUF and ISC protein to interrogate the origin of plastids and mitochondria, and most can be refuted:

1) "they carry out essential functions and are taxonomically well distributed": 

While this is true in eukaryotes, the ISC system is absent from large alphaproteobacterial clades, as seen in Fig 1 and in the text "ISC being present mostly in the basal lineages (Magnetococcales, Marine Proteobacteria 1, MC1) as well as in Rickettsiales, while it is absent in most of the other Alphaproteobacteria,". It seems to be a very odd system to use to place eukaryotes relative to alphaproteobacterial then. How would you be able to infer a more nested position of mitochondria within alphaproteobacterial if most lineages don't possess homologues of the studied system? That strongly discredit the validity of such proteins to address this evolutionary question.

2) "their components are encoded in the nucleus, avoiding the problem of compositional biases;" 

This is not the case for SufB which is often plastid encoded in red algae and secondary algae! See for example Janouškovec 2015

3) "being part of a highly integrated process they were likely subjected to similar evolutionary constraints." 

First of all, the authors should explicit how this would improve phylogenetic resolution. Most importantly, the method section mentions that "Markers that were not broadly distributed (IscX in eukaryotes, SufU and SufT in Cyanobacteria) were eliminated. We also discarded HscB and IscU as they did not form clear monophyletic groups in preliminary trees." This is evidence that the different components of these machines are NOT subjected to similar evolutionary constraints.

--- The authors underplay the lack of resolution of theirs results --- 

The authors first highlight the congruence of their results with what's been inferred from the most sophisticated phylogenomic analyses on the question:

"Bayesian analysis using the CAT+GTR model robustly supports the monophyly of Eukaryotes (PP=1) and their deep branching just after Magnetococcales (PP=0.82), strengthening the Alphaproteobacteria-deep hypothesis4,5" while, in fact, the two cited papers (4,5) recover a more nested position of eukaryotes! Why are the authors not mentioning this incongruence? 

Similarly, for the SUF analyses, they write that "Bayesian and ML analysis supports a deep placement of plastids within Cyanobacteria as also indicated by previous analyses9,10." while in fact the Bayesian phylogeny is almost completely unresolved when it comes to placement of plastids.

Finally, in that analysis, the different approaches yield three different placements for plastids that are listed without much discussion at all or argument for how to explain this or which is most likely the correct placement. 

So not only was there no controversy about the placement of plastids in cyanobacteria in the literature before this work, contrary to the framing of the authors, but here the results do not bring any further insights into it, they're simply inconclusive. 

CONCLUSION:

In conclusion, this work tries to address non-controversial questions with less sophisticated analyses and inconclusive results, while misrepresenting the state-of-the-art and the relevance of their approach. I do not think this is up to the standards of publication in Plos Biology.

---

## [Editor Report · Decision Letter 2]

18 Sep 2023

Dear Simonetta,

Thank you for your patience while we considered your revised manuscript "Phylogenomic analysis of Iron-Sulfur cluster biosynthesis machineries supports a deep origin of mitochondria and plastids" for publication as a Short Reports at PLOS Biology. This revised version of your manuscript has been evaluated by the PLOS Biology editors and the Academic Editor (please note that, as advised by email, this Academic Editor was not involved in the initial decision to review).

Based on our Academic Editor's assessment of your revision, we are likely to accept this manuscript for publication, provided you satisfactorily address the following data and other policy-related requests.

IMPORTANT - Please attend to the following:

a) Please could you change the Title to something like "Phylogenomic analysis of iron-sulfur cluster biosynthesis machineries supports a deep origin of mitochondria and plastids during the diversification of their respective bacterial clades" (at the moment it's unclear what "deep origin" means, so a reader might assume that it refers to deep within Eukaryota, which would be rather unsurprising...)

b) Many thanks for providing the alignments, treefiles, etc in the zipped folder "Garcia_et_al_Supplementary-data.zip" Please could you re-name this "S1 Data"? Please then cite the location of the data clearly in all relevant main and supplementary Figure legends, e.g. “The data underlying this Figure can be found in S1 Data.”

c) Please make any custom code available, either as a supplementary file or as part of a permanent DOI'd deposition.

We expect to receive your revised manuscript within two weeks. 

*Published Peer Review History*

*Press*

Sincerely,

Roli

Roland Roberts, PhD

Senior Editor,

rroberts@plos.org,

PLOS Biology

DATA POLICY:

[Figs….]

CODE POLICY

Per journal policy, as the code that you have generated is important to support the conclusions of your manuscript, we require that you make it available without restrictions upon publication. Please ensure that the code is sufficiently well documented and reusable, and that your Data Statement in the Editorial Manager submission system accurately describes where your code can be found.

DATA NOT SHOWN?

---

## [Editor Report · Decision Letter 3]

10 Oct 2023

Dear Simonetta,

Thank you for the submission of your revised Short Report "Components of Iron-sulfur cluster assembly machineries are robust phylogenetic markers to trace the origin of mitochondria and chloroplasts" for publication in PLOS Biology. On behalf of my colleagues and the Academic Editor, Filipa Sousa, I'm pleased to say that we can in principle accept your manuscript for publication, provided you address any remaining formatting and reporting issues. These will be detailed in an email you should receive within 2-3 business days from our colleagues in the journal operations team; no action is required from you until then. Please note that we will not be able to formally accept your manuscript and schedule it for publication until you have completed any requested changes.

Sincerely,

Roli

Senior Editor

PLOS Biology

rroberts@plos.org